# Out-of-pocket costs for families and people living with cerebral palsy in Australia

**Georgina Henry**[1]*, **Annabel Webb**[1], **Claire Galea**[1], **Alison Pearce**[2,3], **Isabelle Balde**[1], **Fiona Garrity**[4], **Sophie Marmont**[4], **James Espie**[4], **Nadia Badawi**[1,5], **Sarah McIntyre**[1]

**1** Cerebral Palsy Alliance Research Institute, Specialty of Child and Adolescent Health, The University of Sydney, Sydney, New South Wales, Australia, **2** The Daffodil Centre, The University of Sydney, A Joint Venture with Cancer Council NSW, Sydney, New South Wales, Australia, **3** Sydney School of Public Health, The University of Sydney, Sydney, New South Wales, Australia, **4** CP Quest, Cerebral Palsy Alliance, Sydney, New South Wales, Australia, **5** Grace Centre for Newborn Care, Children's Hospital at Westmead, Sydney, New South Wales, Australia

* georgina.henry@cerebralpalsy.org.au

**Data Availability Statement:** Participants did not consent to making their data publicly available and the raw data of this study includes sensitive information and indirect identifiers which may risk the identification of participants. All data requests

## Abstract

The most recent cost estimates of cerebral palsy (CP) in Australia did not include out-of-pocket costs for families. This study aimed to: 1) describe and estimate out-of-pocket costs for people with CP and their families by age and gross motor function classification system (GMFCS) level; 2) measure financial distress. A cross-sectional quantitative survey design was used with qualitative approaches to analyse open-ended questions. A CP-specific out-of-pocket costs survey was co-designed with people with lived experience. Adults with CP and carers were recruited from Australian population-based CP Registers and via social media. Sociodemographic variables were analysed descriptively and median (IQR) expenses for health, assistive technology, personal care, housing, occupation, transport, leisure, respite and holidays, by age (0–6; 7–17; 18 years +) and gross motor function [GMFCS level I-II vs III-V] were calculated. The In Charge Financial Distress/Financial Well-being Scale measured financial distress. Regression analyses were conducted to investigate costs and financial distress. Additional out-of-pocket costs itemised in open-ended questions were charted. Comments were thematically analysed using the framework approach. 271 surveys were completed for children 0–6 years (n = 47), children/adolescents 7–17 years (n = 124) and adults (n = 100). 94% of participants had out-of-pocket costs associated with CP, with an overall annual median of $4,460 Australian dollars (IQR $11,955). After controlling for income, private insurance and disability funding, the GMFCS III-V group had costs two times higher than the GMFCS I-II group (2.01; 95% CI 1.15–3.51). Age was not significantly associated with costs. 36% of participants had high to overwhelming financial distress; this was not associated with age or GMFCS level after controlling for financial factors. Families had several additional disability costs. Open-ended responses revealed experiences of financial concern were influenced by funding scheme experiences, reduced income, uncertainty, access to support networks and an inability to afford CP-related costs. Cost estimates and financial distress indicators should inform policy, funding and clinical decisions when planning interventions to support people with CP and their families.

can be sent to The University of Sydney Human Research Ethics Committee human. ethics@sydney.edu.au.

**Funding:** A University of Sydney Faculty of Medicine and Health Ignition Seed Fund Grant was awarded to AP and was used for participant gift vouchers and the preparation of dissemination materials. The funders had no role in study design, data collection and analysis, decision to publish or preparation of the manuscript. https://www. sydney.edu.au/medicine-health/.

**Competing interests:** The authors have declared that no competing interests exist.

## Introduction

Cerebral palsy (CP) is an umbrella term for a group of permanent movement and/or posture conditions, caused by an injury to the developing brain. Individuals with CP may also have comorbidities such as epilepsy, intellectual disability, hearing and vision impairments [1]. Cerebral palsy can have a range of impacts on individuals, with some people requiring high levels of support for daily activities. The lifelong management of CP can pose significant financial costs to families and the economy [2].

A systematic review examining the costs of CP worldwide estimated that medical expenses for children with CP were 10–26 times higher than typically developing children, with a positive relationship between gross motor severity and expenses [3]. The most recent cost estimate in Australia of CP determined a cost of $145, 632 AUD per person per annum [2]. This encompassed a range of costs including productivity loss for the person with CP and carer, wellbeing loss, disability support and other costs. However, it did not include out-of-pocket financial costs for families and individuals with CP. Out-of-pocket costs refer to financial expenses which families and individuals pay for using their own income [4]. These expenses are typically either not covered, or only partly covered, by government funding or insurance schemes. One study from Australia estimated out-of-pocket costs associated with early intervention programs for families. Throughout this period families spent a median of $57, 000 AUD on equipment and considerable costs were associated with accessing health and non-mainstream interventions (e.g. Chinese medicine) [4, 5]. Meeting these costs often came at the expense of other goods and experiences (e.g. clothing, holidays). This study was conducted prior to the completed roll-out of the National Disability Insurance Scheme (NDIS)–a national funding scheme in Australia designed to provide "necessary and reasonable supports" to people with disability and families [6]. By allocating government funding directly to individuals and families, NDIS funding may significantly reduce out-of-pocket costs.

Out-of-pocket costs related to health may cause financial distress. Financial hardship has been measured in other populations who experience chronic health conditions [7, 8]. Whilst parents of children with CP may experience financial hardship and distress, this is yet to be examined in the context of a national disability funding scheme [9–11].

This study aims to: 1) describe and estimate the out-of-pocket costs for individuals with CP and their families in Australia by age and gross motor function severity; and 2) measure the financial distress experienced by individuals with CP and/or their families.

## Methods

This study was conducted in partnership with three Research Partners with lived experience from CP Quest https://cerebralpalsy.org.au/our-research/get-involved-research/cp-quest/. The partners are co-authors, and include an adult with CP (S.M.), a parent of a child with CP (J.E.) and parent of a young adult with CP (F.G.). The reporting of this study follows the Consensus-Based Checklist for Reporting of Survey Studies (CROSS) [12; see S1 Table].

Cross-sectional survey questions were drafted based on the Client Service Receipt Inventory (CSRI), which had been previously modified to measure costs associated with intellectual disability in Australia [13]. Research partners were consulted to ensure included questions addressed CP-specific costs. Based on this feedback, the survey was developed and modified through an iterative process with the research partners to ensure relevance of items, ease of use and acceptability.

The final internet-based survey comprised questions addressing demographics, clinical details, and disability expenses across nine cost types (health, assistive technology, personal care, housing, occupation, transport, leisure, respite and holidays). Participants were asked to

estimate their out-of-pocket costs "over the last 12 months" for most cost types. Items addressing the cost of health services were framed as "per visit" to minimise recall bias. "Over the last 5 years" was used for the larger, more infrequent expenses: assistive equipment, transport and home modifications. Financial distress was measured using the reliable and valid In Charge Financial Distress/Financial Well-Being Scale [IFDFW; 14]. IFDFW scores range from 1 to 7, with lower scores indicating higher levels of financial distress. The entire online survey included 144 questions, with a maximum of 6 questions per page across 36 pages. Items consisted of yes and no, multiple choice, and open-ended questions. Non-response options (e.g. "I'd rather not say") were included for questions pertaining to sensitive information (e.g. income); these questions were not compulsory. The survey was available in English and was designed for either an adult with CP or a carer to complete (see S1 Appendix).

People with CP and families living in Australia were invited to participate through a snowballing recruitment strategy. Three state CP Registers sent an email to families who consented to receive research invitations. The CP Registers are a reflective sample of the population of people with CP in Australia. A study advertisement was included in the bi-annual NSW/ACT CP Register Newsletter and distributed through Cerebral Palsy Alliance's social media and websites. Advisory groups and collaborators were asked to forward the invitation through their networks. There was no available data to base a sample size calculation on. Based on clinical knowledge of the field, and for generalisability of results, the team identified a prior that 180 participants were required across three age and two severity brackets to ensure a range of responses, reflecting the CP population.

The open, voluntary survey introduction included the participant information sheet and outlined the survey purpose, who the investigators were, how the information would be stored, and the length of time it would take to complete the survey (30–90 minutes). Participants were required to tick "I have read the participant information sheet" before they commenced the survey. The survey was open from August 2020 to Feb 2021. Participants were able to stop the survey at any time, or review and change their responses over this time period. Responses were automatically captured by REDCap (REDCap 11.0.3) [15, 16]. No identifiable information was collected, however at survey completion participants had the option to leave their email address to receive updates and/or to have a chance to win a $50 voucher. The researchers conducting data analysis had no access to any identifiable information. An IP address check was conducted to ensure only one entry per person.

### Ethics statement

The study was approved by The University of Sydney Human Research Ethics Committee (2020/234). Written consent was not required; informed consent was implied following voluntary commencement of the survey.

### Analysis

Demographic and clinical characteristics were summarised using medians and inter-quartile ranges (IQR) for continuous variables and frequencies for categorical variables. The median and IQR of out-of-pocket costs, and the proportion of respondents reporting no out-of-pocket costs, were computed. Total annual out-of-pocket costs were determined by summing all reported annual costs. Where required, reported out-of-pocket costs were annualised or calculated using reported costs for time scales other than 12 months. Out-of-pocket costs were stratified by age groups and gross motor function classification system (GMFCS) levels. GMFCS I-II levels were combined (can walk independently), and GMFCS III-V levels were combined (requires mobility assistance). The age groups 0–6, 7–17, 18+ were chosen to resemble NDIS

funding categories. Out-of-pocket costs individually itemised by participants in open-ended questions, which were not explicitly asked about in the survey, were charted and summarised, but not included in calculations.

Multiple regression was used to identify whether there was a significant association between total out-of-pocket costs and age group and/or GMFCS levels. Logistic regression was used to investigate associations with the odds of having no out-of-pocket costs. Gamma regression was used to investigate associations with total out-of-pocket costs to account for the substantially right-skewed distribution of responses. Regression models were adjusted for private health insurance status, household income and disability funding amount (purposefully selected) to control for their confounding effect on expenditure. Similar regression models were used to investigate whether age group or GMFCS level was associated with out-of-pocket costs for specific cost types. Analyses concerning specific types of costs only included participants who reported the amount spent on the relevant cost type. Possible association between disability funding amount, household income and private health insurance status was investigated using ANOVA models. The percentage of participants who scored between 1 and 4 on the IFDFW (high to overwhelming levels of distress) was calculated. Linear regression was used to identify whether there were significant associations between age group or GMFCS level and financial distress levels. All analyses were performed in R v4.1.0 [17]. A p-value of <0.05 was considered statistically significant for all analyses.

Responses to open-ended questions complimented the quantitative data, and assisted with the interpretation and explanation of cost estimates (aim 1) and financial distress (aim 2) scores [18]. Comments were thematically analysed inductively using the Framework approach [19–21]. Two researchers (GH and SMc) independently read the open-ended text responses and a coding framework was formulated. The framework was systematically applied to all open-ended responses and inconsistencies were discussed. Coded responses were then synthesized into a set of thematic matrix charts to refine overarching themes. Researcher triangulation was adopted to increase the trustworthiness of the qualitative analysis–all analyses were reviewed and discussed among our multi-disciplinary team of investigators, including our research partners with lived experience of CP. Participants across all subgroups (age and GMFCS levels) responded to open-ended questions, indicating the case-to-case transferability of the data [22].

## Results

### Demographics

A total of 346 surveys were commenced and 271 were completed, resulting in a completion rate of 78%. Open-ended questions were completed by 231 (85%) participants. Completed surveys were included in all analyses and comprised data for children 0–6 years (n = 47), children/adolescents 7–17 years (n = 124) and adults (n = 100). All types and severity of CP (49% GMFCS I-II; 44% GMFCS III-V; 7% missing/unknown) were represented and 85% of participants reported at least one comorbidity (Table 1).

Forty-one percent of adult survey responses were self-reported. Adults with CP predominantly (43%) lived at home with two parents, 9% had full-time work and half were either a student (30%) or engaged in part-time employment (20%). Of the adults who completed the survey via self-report 54% indicated they earned less than $31,199 per year, and 10% reported earning more than $104, 000 per year. In comparison, 14% of carers reported a household income of less than $31,199 per year and 43% a household income of more than $104, 000 per year. The majority (85%) of participants had accessed disability funding in the last 12 months and half had private health insurance. There was no relationship between the amount of

**Table 1. Clinical and sociodemographic characteristics.**

| Clinical/Sociodemographic Characteristic | Age 0–6 n (%) | Age 7–17 n (%) | Age 18+ n (%) |
|---|---|---|---|
| Total n | 47 | 124 | 100 |
| Gender of Person with CP | | | |
| Female | 14 (30) | 45 (36) | 61 (61) |
| Male | 33 (70) | 79 (64) | 36 (36) |
| Neutral/ Prefer not to disclose | 0 (0) | 0 (0) | 3 (3) |
| Cerebral Palsy Motor Type(s) | | | |
| Spastic | 32 (68) | 94 (78) | 69 (78) |
| Ataxic | 7 (15) | 13 (11) | 17 (19) |
| Dyskinetic | 12 (26) | 22 (18) | 13 (15) |
| Unknown | 0 (0) | 5 (4) | 2 (2) |
| GMFCS Level | | | |
| Level I | 13 (28) | 24 (19) | 15 (15) |
| Level II | 15 (32) | 40 (32) | 27 (27) |
| Level III | 5 (11) | 10 (8) | 15 (15) |
| Level IV | 4 (9) | 20 (16) | 13 (13) |
| Level V | 8 (17) | 26 (21) | 18 (18) |
| Unknown | 2 (4) | 1 (1) | 0 (0) |
| Missing | 0 (0) | 3 (2) | 12 (2) |
| Level I-II | 28 (62) | 64 (53) | 42 (48) |
| Level III-V | 17 (38) | 56 (47) | 46 (52) |
| Other Impairments | | | |
| No Other Impairment | 14 (30) | 13 (10) | 14 (14) |
| Epilepsy | 10 (21) | 48 (39) | 29 (29) |
| Intellectual Impairment | 8 (17) | 59 (48) | 33 (33) |
| Visual Impairment | 11 (23) | 42 (34) | 25 (25) |
| Hearing Impairment | 6 (13) | 7 (6) | 6 (6) |
| Speech Impairment | 25 (53) | 62 (50) | 35 (35) |
| ADHD | 4 (9) | 18 (15) | 6 (6) |
| ASD | 6 (13) | 19 (15) | 9 (9) |
| Mental Health Condition | | | 8 (8) |
| Other Impairment | 5 (11) | 17 (14) | 11 (11) |
| Disability Funding | | | |
| Accessed disability funding in the previous 12 months | 45 (96) | 107 (88) | 77 (81) |
| Median Annual Disability Funding Amount (IQR) | $28, 000.00 ($21, 000.00) | $27, 500.00 ($75, 750.00) | $50, 000 ($120, 000.00) |
| Health Insurance Status | | | |
| Private Insurance | 24 (51) | 58 (48) | 58 (60) |
| Receive bulk billing | 24 (51) | 60 (50) | 37 (39) |
| Other | 20 (43) | 55 (45) | 31 (32) |
| Highest Level of Education of Carer (carer report only) | | | |
| Primary Education | 0 (0) | 1 (1) | 3 (5) |
| Secondary Education | 5 (11) | 19 (15) | 21 (36) |
| Vocational Training/Diploma | 13 (28) | 32 (26) | 11 (19) |
| Tertiary Education | 29 (62) | 68 (55) | 24 (41) |
| Other/None of the above | 0 (0) | 3 (2) | 0 (0) |
| Missing/Unknown | 0 (0) | 1 (1) | 0 (0) |
| Household income before tax (carer report only) | | | |
| Less than $31,199 per year | 6 (13) | 18 (15) | 7 (13) |

*(Continued)*

**Table 1.** (Continued)

| Clinical/Sociodemographic Characteristic | Age 0–6 n (%) | Age 7–17 n (%) | Age 18+ n (%) |
|---|---|---|---|
| $31,200 to $51,999 per year | 3 (6) | 18 (15) | 13 (24) |
| $52,000 to $103,999 per year | 9 (19) | 27 (22) | 10 (18) |
| More than $104,000 per year | 27 (57) | 54 (45) | 16 (29) |
| Did not wish to disclose | 2 (4) | 6 (5) | 9 (16) |
| Missing/Unknown | 0 (0) | 1 (1) | 0 (0) |

disability funding received and household income (p = 0.861) or private health insurance (p = 0.244). Clinical and sociodemographic information is summarized in Table 1 and S2 Table.

## Overall out-of-pocket costs

Ninety-four percent of participants had out-of-pocket costs associated with CP. There were no significant clinical or demographic differences between those who had out-of-pocket costs and those who did not (not reported). For the few participants who reported no out-of-pocket costs (n = 13), the majority (77%) had CP classified as GMFCS I-II. Due to the small number of participants in this group there were no statistically significant associations detected between having zero out-of-pocket costs and GMFCS level or age group. Therefore, only participants who reported costs (n = 214) were included in analyses concerning overall out-of-pocket costs. The overall median out-of-pocket-costs for people with CP and their carers was $4,460 (IQR $11,966) per year, however 20% of participants had annual expenses over $20,000 per year (median = $38,020; IQR $26, 820).

## Costs by GMFCS level and age

After adjusting for age, income, private insurance and disability funding, overall out-of-pocket costs were two times higher on average for the GMFCS III-V group compared to GMFCS I-II group (2.01; 95% CI 1.15–3.51). Overall, age was not significantly associated with out-of-pocket cost amount. Relative to the adult group, costs for young children (0–6 years) were 8% lower on average (ratio 0.92; 95% CI: 0.43, 1.98), and costs for children/adolescents (7–17 years) were 6% lower on average (ratio 0.94; 95% CI: 0.49, 1.78).

Across all age groups, the annual median out-of-pocket costs for individuals with CP classified as GMFCS I-II was $2,660 (IQR $5280). Medians for the GMFCS I-II groups ranged from $1, 632 (IQR $6, 620) per year for young children (0–6 years) to $2, 900 (IQR $8, 404) for adults (Fig 1). Across all age groups, the annual median out-of-pocket costs for individuals with CP classified as GMFCS III-V was $10, 900 (IQR $18, 065). For the GMFCS III-V group, median annual out-of-pocket costs for young children (0–6 years) was $3, 546 (IQR $7, 074), increasing to $12, 345 (IQR $30, 951) for children/ adolescents (7–17 years) (Fig 1). This was the largest increase across age groups, with adults in the GMFCS III-IV group having a median of $11, 828 (IQR $12, 968). Across all GMFCS levels and age groups, some families had substantially increased costs as seen in Fig 1.

## Types of costs by GMFCS level and age

Participants reported having a range of out-of-pocket costs across each of the cost types regardless of GMFCS level and age. However, the amount of money spent on specific cost

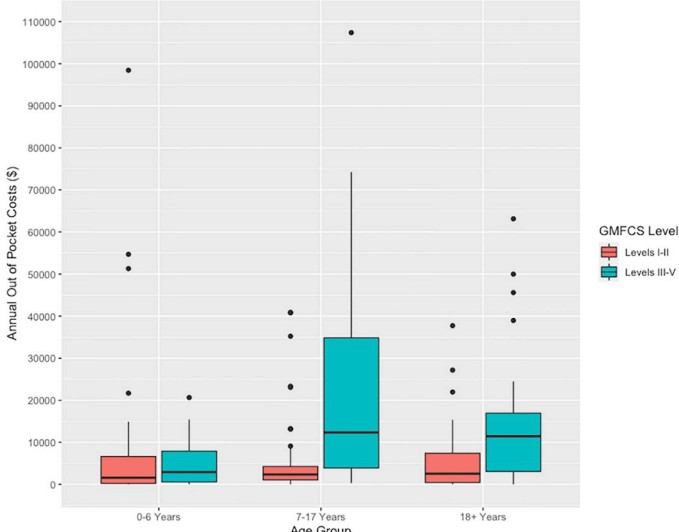

**Fig 1. Annual out-of-pocket costs by GMFCS level and age group.**

types differed depending upon GMFCS level and age. After adjusting for age and income, the GMFCS III-V group had significantly greater out-of-pocket costs for assistive technology (p<0.001) and transport (p<0.001) relative to the GMFCS I-II group. After adjusting for GMFCS level and income, age was significantly associated with out-of-pocket home modification costs (p = 0.03). Costs of other types did not significantly differ between age groups or GMFCS level groups after adjusting for income.

The largest median annual cost for the GMFCS I-II group was $1,085 (IQR $2, 662) for health costs, whilst for the GMFCS III-V group it was $4,700 (IQR $11, 025) for transport costs. Table 2 further outlines median annual costs across cost types by age and GMFCS level. A comprehensive breakdown of median costs for thirty different cost subtypes (e.g. neurologist, allied health, rehabilitation specialist) by age group can be seen in Table 3. There were no predominant drivers of out-of-pocket costs, instead all subtypes were represented, showing a breadth of costs for families. Regression analyses are summarised in Table 4.

**Additional costs.** Open-ended responses indicated families had numerous additional costs which were not explicitly asked about in the survey. Participants identified additional costs associated with health, assistive technology, personal care and housing. Whilst some participants paid for these additional expenses, others commented they were unable to (Table 5).

Several participants highlighted the significant cost of time and emotional impact of caring for a family member with CP, distinct from financial distress.

*"I am disappointed that this survey about 'costs' does not identify any financial value of carer time, and their lost opportunities in life"*

*(P51).*

*"[This survey] does not factor in the financial cost of emotional stress placed on parents and carers"*

*(P197).*

**Table 2. Annual costs across cost type.**

| Type of Cost (n = responders) | GMFCS Level | Median (IQR$) Annual Costs | | | |
|---|---|---|---|---|---|
| | | Age 0–6 years | Age 7–17 years | Age 18+ years | All Ages |
| Health *n = 178* | GMFCS I-II | $840 (4205) | $1264 (2320) | $882* (3141) | $1085 (2662) |
| | GMFCS III-V | $225* (852) | $1600 (3339) | $1350* (2800) | $1350* (2788) |
| Mobility/Comm Tech *n = 148* | GMFCS I-II | $500 (1575) | $200 (300) | $200 (310) | $200 (400) |
| | GMFCS III-V | $120* (100) | $900 (4600) | $380 (3775) | $540 (4480) |
| Clothing, Nutrition and Personal Care *n = 135* | GMFCS I-II | $1000 (2000) | $330 (714) | $500 (1900) | $500 (1250) |
| | GMFCS III-V | $1000 (1550) | $1200 (2400) | $400 (850) | $700 (1700) |
| Home *n = 127* | GMFCS I-II | $200 (1715) | $200 (1290) | $500 (963) | $200 (1210) |
| | GMFCS III-V | $400 (1440) | $4000 (19500) | $1100 (2580) | $1350 (4950) |
| Occupation/ Education *n = 73* | GMFCS I-II | $100 (475) | $1300 (2275) | $400 (1500) | $750 (1975) |
| | GMFCS III-V | $1750 (1650) | $400 (800) | $800 (2025) | $500 (1700) |
| Transport *n = 170* | GMFCS I-II | $530 ($1300) | $200 ($765) | $1000 ($1500) | $500 ($1100) |
| | GMFCS III-V | $1250 ($5638) | $7480 ($12345) | $4400 ($8105) | $4700 ($11025) |
| Leisure *n = 87* | GMFCS I-II | $1200 (1500) | $500 (650) | $1500 (2010) | $500 (1038) |
| | GMFCS III-V | $400 (450) | $750 (800) | $600 (283) | $600 (763) |
| Respite *n = 46* | GMFCS I-II | - | $800 (-) | - | $800 (-) |
| | GMFCS III-V | - | $825 (463) | $700 (400) | $825 (613) |
| Holidays *n = 16* | GMFCS I-II | $1000* (-) | $250* (505) | $1800 (2900) | $600* (800) |
| | GMFCS III-V | - | $400* (335) | $3000* (4000) | $1000* (3550) |

* Between 20–35% of people reported using this service/ product but did not indicate whether they had associated out-of-pocket expenses

## Desire to spend more and financial distress

More than half the participants reported they would like to spend more money on leisure (56%) and holidays (65%) if they were able. Over a third (38%) also reported they would like to spend more on health-related costs. For individuals classified as GMFCS III-V, more than two-thirds (69%) would spend more money on home modifications, and a half (52%) on assistive technology. S3 Table outlines the percentage of participants by age and GMFCS grouping who desired to spend more money across the nine cost types.

One in three (36%) participants experienced high to overwhelming levels of financial distress (Overall $M$ = 5.112; $SD$ = 2.315). After accounting for income, private insurance and disability funding, neither age (p = 0.336) nor GMFCS level (p = 0.088) were significantly associated with financial distress level.

Qualitative analysis revealed two overarching themes which further explained participants' experiences of financial concern and desire to spend more: i) Causes and inhibitors of distress; and ii) Lack of out-of-pocket expenses—going without

**Causes and inhibitors of financial concern.** Analysis of open-ended responses revealed four factors which may cause or inhibit experiences of financial concern: i) Funding scheme experiences; ii) Reduced income; iii) Uncertainty; and iv) Support networks.

There were varied responses regarding experiences of the NDIS. Many participants emphasized that their financial concern had improved since the introduction of the NDIS. Others found accessing the NDIS a very difficult and frustrating process. Some participants expressed their gratitude for NDIS support but acknowledged that it did not meet all their financial disability-related needs.

Many participants highlighted that a major source of financial concern was having a reduced income. Some participants were unable to work due to caring responsibilities, whilst

**Table 3. Annual costs across sub-types.**

| Cost Type | Out-of-Pocket Costs | | |
|---|---|---|---|
| **Health** | | | |
| | **Age 0–6 years** | **Age 7–17 years** | **Age 18 + years** |
| | Median (IQR$) costs per visit to health service | | |
| General practitioner | $38 (35) | $50 (41) | $40 (19) |
| Neurologist | $300 (75) | $164 (240) | $165 (93) |
| Rehab service | $150 (105) | $50 (95) | $90 (140) |
| Specialist | $160 (121) | $130 (160) | $150 (100) |
| Allied health (Physiotherapy, Occupational and Speech therapy) | $150 (261) | $80 (130) | $100 (85) |
| Other allied health | $100 (490) | $110 (117) | $100 (132) |
| Alternative treatments | $15 (17) | $50 (113) | $85 (58) |
| | Median (IQR$) costs over 12 months | | |
| Medication | $200 (375) | $400 (850) | $200 (423) |
| Hospitalisation | $500 (1725) | $300 (800) | $620 (2488) |
| Mobility/ Communication Technology | | | |
| | Median (IQR$) costs over 5 years | | |
| Communication equipment | $200 (1225) | $3000 (3500) | $1000 (14000) |
| Mobility equipment | $750 (4325) | $1500 (4850) | $275 (1575) |
| Clothing, Nutrition and Personal Care | | | |
| | Median (IQR$) costs over 12 months | | |
| Specialised Clothing | $1500 (1400) | $500 (400) | $400 (575) |
| Specialised Nutrition | $500 (1100) | $1000 (1625) | $550 (1175) |
| Specialised Personal Care | $600 (500) | $500 (1300) | $150 (287.5) |
| Housing | | | |
| Costs of moving location | $30,000 (417, 500) | $32, 500 (174, 750) | $10, 000 (122, 600) |
| | Median (IQR$) costs over 5 years | | |
| Home access modifications | $5000 (6250) | $7000 (48000) | $4500 (8000) |
| Internal home modifications | $1500 (8125) | $13500 (32125) | $5000 (12500) |
| Home equipment | $500 (720) | $2000 (4200) | $500 (2800) |
| Occupation/Education | | | |
| | Median (IQR$) costs over 12 months | | |
| Occupation/education modifications | $525 (475) | $275 (1125) | $1000 (950) |
| | Median (IQR$) costs over 5 years | | |
| Other occupation/education costs e.g. tutor, support aides, private school fees | $500 (8250) | $2750 (7625) | $2000 (5250) |
| Transport | | | |
| | Median (IQR$) costs over 12 months | | |
| Public transport | $75 (50) | $100 (150) | $175 (440) |
| Tolls/parking | $200 (410) | $200 (400) | $200 (400) |
| Other transport costs e.g. taxis, parking fines during hospital visits running late | $400 (700) | $300 (300) | $225 (475) |
| | Median (IQR$) costs over 5 years | | |
| Car purchase | $35000 (25000) | $40000 (35000) | $26500 (25125) |
| Car modifications | $6000 (0) | $3750 (25500) | $2500 (13000) |
| Other car- related costs e.g. insurance, repairs, petrol, wear and tear, car hire | $4800 (2300) | $3000 (3500) | $5000 (15000) |
| Leisure, Respite Care and Holidays | | | |
| | Median (IQR$) costs over 12 months | | |
| Leisure or sport | $850 (1025) | $500 (800) | $550 (1378) |
| Other leisure or sports costs (e.g. modified music or sporting equipment) | $1000 (0) | $300 (390) | $6 (4) |
| Respite care | - | $800 (350) | $700 (400) |
| Holidays (e.g. accessible accommodation, support worker, equipment hire) | $1000 (0) | $300 (390) | $2000 (4100) |

**Table 4. Cost type regression analyses.**

| Cost Type | Adjusted effect, exp(beta) (95% CI) | p-value |
|---|---|---|
| Overall | | |
| Age 0–6 (vs Age 18+) | 0.92 (0.43, 1.98) | 0.829 |
| Age 7–17 (vs Age 18+) | 0.94 (0.49, 1.78) | 0.870 |
| GMFCS III-V (vs GMFCS I-II) | **2.01 (1.15, 3.51)** | **0.016** |
| Health | | |
| Age 0–6 (vs Age 18+) | 1.51 (0.48, 4.63) | 0.433 |
| Age 7–17 (vs Age 18+) | 0.88 (0.36, 1.96) | 0.764 |
| GMFCS III-V (vs GMFCS I-II) | 1.16 (0.57, 2.35) | 0.642 |
| Mobility/Communication Tech | | |
| Age 0–6 (vs Age 18+) | 6.33 (0.45, 32.16) | 0.405 |
| Age 7–17 (vs Age 18+) | 1.99 (0.81, 4.44) | 0.112 |
| GMFCS III-V (vs GMFCS I-II) | **7.07 (3.02, 15.71)** | **<0.001** |
| Clothes, Nutrition & Personal Care | | |
| Age 0–6 (vs Age 18+) | 0.77 (0.19, 3.40) | 0.720 |
| Age 7–17 (vs Age 18+) | 1.15 (0.37, 3.06) | 0.798 |
| GMFCS III-V (vs GMFCS I-II) | 0.63 (0.27, 1.51) | 0.292 |
| Home | | |
| Age 0–6 (vs Age 18+) | 1.90 (0.47, 7.28) | 0.308 |
| Age 7–17 (vs Age 18+) | **4.21 (1.67, 9.98)** | **0.030** |
| GMFCS III-V (vs GMFCS I-II) | 2.34 (0.80, 5.91) | 0.087 |
| Occupation/Education | | |
| Age 0–6 (vs Age 18+) | 0.23 (0.03, 2.31) | 0.148 |
| Age 7–17 (vs Age 18+) | 0.36 (0.08, 1.41) | 0.116 |
| GMFCS III-V (vs GMFCS I-II) | 0.54 (0.16, 1.82) | 0.253 |
| Transport | | |
| Age 0–6 (vs Age 18+) | 0.78 (0.34, 1.72) | 0.493 |
| Age 7–17 (vs Age 18+) | 0.66 (0.34, 1.22) | 0.162 |
| GMFCS III-V (vs GMFCS I-II) | **2.08 (1.29, 3.23)** | **0.002** |
| Leisure | | |
| Age 0–6 (vs Age 18+) | 0.40 (0.04, 3.42) | 0.267 |
| Age 7–17 (vs Age 18+) | 0.26 (0.05, 1.31) | 0.359 |
| GMFCS III-V (vs GMFCS I-II) | 0.67 (0.26, 1.75) | 0.405 |
| Respite Care | | |
| Age 0–6 (vs Age 18+)[a] | - | - |
| Age 7–17 (vs Age 18+) | 0.80 (0.47, 1.32) | 0.493 |
| GMFCS III-V (vs GMFCS I-II) | 0.25 (0.13, 1.04) | *0.051* |
| Holidays | | |
| Age 0–6 (vs Age 18+) | 0.19 (0.01, 21.43) | 0.328 |
| Age 7–17 (vs Age 18+) | 0.21 (0.04, 9.60) | 0.387 |
| GMFCS III-V (vs GMFCS I-II) | 0.59 (0.13, 3.07) | 0.431 |

[a]: No respondents in this category reported any out-of-pocket costs for respite

others had reduced working hours. This worry was exacerbated by uncertainty surrounding securing adequate disability funding, incurring unexpected and/or changing costs, and a lack of financial security and saving ability. Financial support from friends and family networks assisted in reducing financial concern. Illustrative quotes can be seen in Table 6.

**Table 5. Additional costs identified by participants in extension and general open-ended questions.**

| Cost Type | Additional Cost | Spent | Would if could |
|---|---|---|---|
| Health | | | |
| | Food at hospital visits (appointments running late/hospital stays) | x | |
| | Private health insurance with ambulance cover | x | |
| | Childcare for siblings when attending hospital appointments/ stays | x | |
| | Treatment of chronic injuries/mental health of carers | x | x |
| | Accommodation for caregivers during hospital stays | x | |
| | More specialised toys and equipment for home therapy/ post-surgery intervention | | x |
| | Service animal to assist with emotional support needs | x | |
| Assistive Technology | | | |
| | Trialling devices before purchasing/ordering | x | x |
| | Upgrading and repairing iPads/software/apps | x | x |
| | Upgrading and repairing wheelchairs, seating, prams | x | x |
| Clothing, Nutrition and Personal Care | | | |
| | Buying multiple pairs of shoes or clothes (to accommodate AFOs or to maintain cleanliness) | x | |
| | Increased amount of cleaning products, clothing/table protectors | x | |
| | Increased necessity for washing and drying laundry | x | |
| Home Costs | | | |
| | Increased rent associated with needing a bigger house for accessibility | x | x |
| | Repairs to house walls (e.g. due to equipment/ behavioural issues) | | x |
| | Increased home contents insurance due to value of assistive technology | x | |

**Lack of out-of-pocket expenses—Going without.** Many participants explained they were often unable to afford disability-related goods and services and would go without them. For example, some participants commented they were unable to afford to buy a new car, and so were ineligible to receive funding for vehicle modifications. Similarly, participants who lived in rental accommodation were also unable to access funding for permanent home modifications.

Participants also explained they often were unable to afford to spend money on holidays or leisure.

"*The carers payment and allowance is not enough to live on, let alone save any for going out, movies, dinners or holidays*"

(P132)

## Discussion

This study was the first to describe and estimate out-of-pocket costs and financial distress experienced by individuals with CP and their families, across age groups and GMFCS levels. Most participants reported having substantial out-of-pocket costs across a range of different cost types, regardless of age and GMFCS level. Overall, out-of-pocket costs for individuals with CP classified as GMFCS III-V were two times higher than those with milder CP, having greater assistive technology and transport costs. Over a third of participants also reported high to overwhelming levels of financial distress, with many desiring to spend more and reporting going without.

Despite accessing NDIS funding, the majority of participants still incurred substantial out-of-pocket costs because of CP. These costs spanned across nine different cost types and thirty

**Table 6. Factors affecting experiences of financial concern.**

| Factor | Example Quote | Participant Number |
|---|---|---|
| Funding Scheme experiences | | |
| | Access to (the) NDIS has reduced a lot of my financial stress | P74 |
| | Since we started receiving NDIS support it has been much easier, much less out of pocket costs | P179 |
| | The only reason we see therapists as often as we do is because NDIS makes it possible | P209 |
| | NDIS has been life changing and has enabled us to get all the Assistive Technology he needs | P236 |
| | I often can't get the equipment I actually need because the NDIS believes that I should have to fund it myself and they don't believe it's reasonable and necessary | P37 |
| | NDIS does not allow for enough flexibility in funding to cover all things required | P116 |
| | We don't do NDIS because it is such a difficult process | P207 |
| | Funding has been better since (accessing the) NDIS but there are still gaps. Equipment and disability specific items are out of reach without funding, but the process can be long and stressful. | P72 |
| Reduced Income | | |
| | My income has halved since caring for my child with disability | P18 |
| | My earning potential has been significantly reduced due to frequency of early intervention therapies and level of in-home therapy required to support my child. Not only does this result in reduced income, but also impacts on my superannuation. | P66 |
| | Our financial stress has been relieved greatly due to the ability for me to (return to) work. | P147 |
| | We have been a single income family for X+ years, and now pensioners, resulting in a tenuous financial position. But for a modest inheritance X years ago we would be in severe financial stress. | P169 |
| | This also reduces our ability to constantly have savings to fall back on when required due to costs of living | P266 |
| Uncertainty | | |
| | I think not knowing what might come next and how much it might cost with regards to caring for someone with CP is considered on a daily basis | P52 |
| | I am grateful to have so many of my child's needs covered by NDIS but it is a stressful fight to get those funds and not a certainty they will be consistent each year...unexpected medical costs crop up all the time | P65 |
| | I myself have no financial security. as I have cared for her full time for the last X years, my job prospects are low and I am concerned about my future finances and expenses | P72 |
| | I have to be prepared to attend appointments at whatever time I am given by the hospital... It would play havoc with work commitments and be just too difficult to juggle | P217 |
| | We are constantly worried about my husband losing his job as we have no backup | P247 |
| Support Networks | | |
| | If it wasn't for my parents, I would be in a very hard place financially. I am saving as much as I can as are my parents for when they are too old to work or are dead | P23 |
| | I can afford my health care because I live at home, work part-time and am on a disability pension | P26 |
| | Our friends and family were generous enough to donate money so we could pay for it (car modifications) | P32 |
| | We are very fortunate to have family financial assistance | P70 |
| | Without her aunty (my sister) over the last XX years helping to provide for my daughter, I know my daughter wouldn't be where she is today, nor her sister and myself | P195 |

subtypes, indicating that numerous disability-related financial needs contributed to participants' overall out-of-pocket costs. Whilst the NDIS has revolutionised the financial support available to people with disability and families [23, 24], there are often additional hidden costs associated with disability [25]. For example, NDIS funding does not cover health costs (e.g. medication, specialist appointments, hospitalisation), which are often increased for people with a disability. Participants also reported additional challenges accessing funding when experiencing socioeconomic disadvantage, for example those who seemed to be ineligible for home or vehicle modifications because they were unable to afford their own home or a

relatively new car. This finding is reflective of the inverse care law, whereby those who deserve the most support, are least able to access the care they need [26]. Furthermore, whilst the NDIS has significantly enhanced the autonomy individuals and families have to choose the supports and services they receive, several factors can restrict how individuals are able to access and use their NDIS funding. These may include the complexity of NDIS regulations, the range of disability expertise among NDIS planners and awareness of what can and cannot be funded [23, 27]. This highlights the need for all NDIS participants to have an equitable understanding about how funding can be allocated, secured and used.

Access to assistive technology for children and adolescents with severe CP is imperative, providing the basic human right to mobilise, communicate and participate at home, school and within the community [23]. A lack of funding for appropriate assistive technology can potentially lead to out-of-pocket costs for families, as seen in this study. Transport and home modifications were among the highest out-of-pocket costs for families of children with severe CP during early intervention periods, with Bourke-Taylor et al. [4] commenting that these were likely to increase as access requirements change with age. The current finding that age was significantly associated with home modification costs supports this notion, with the greatest costs incurred by adolescents in the GMFCS III-V group. Individuals who experience more severe CP are still incurring significantly greater out-of-pocket costs because of the severity of their disability. This indicates an urgent need to ensure necessary assistive technology, transport and home modifications are equitably available to all families.

Over a third of participants experienced high to overwhelming levels of financial distress, a finding consistent with previous studies indicating financial hardship experienced by individuals living with chronic health conditions [7, 8]. In our study, experiences of financial distress were reportedly influenced by having a reduced income. These results corroborate previous findings that loss of income and associated stress is commonly experienced by families when caring for a person with disability [2, 10, 13, 28]. In addition, only nine percent of adults with CP had full time employment, indicative of barriers to accessible employment opportunities which typically commence during teenage years, greatly reducing income potential [29]. This is reflected in the majority (54%) of adults with CP who self-reported earning less than $31, 199 p.a. This indicates highly disproportionate levels of financial poverty, as compared to 21% of the Australian adult population during 2019–2020 [30]. Participants also commented that uncertainty, experiences engaging with the NDIS and access to support also influenced financial concern. This is consistent with existing qualitative literature which has identified that whilst many carers are grateful for the support of the NDIS, there is much uncertainty around securing funding and navigating the scheme, with many families valuing access to support networks [23, 31].

The potential impact of this financial distress on people living with CP and their carers is worrying. Some families reported being unable to afford disability-related expenses, indicating that participants who had no, or little out-of-pocket costs may still be experiencing financial hardship. Studies in oncology and paediatric disability populations have identified that cost-prohibitive behaviours and financial hardship can adversely affect patient health, resulting in poorer clinical outcomes and a decreased quality of life [32–34]. Financial hardship has also been positively correlated with caregiver burden, detrimentally affecting psychological wellbeing as described by participants in the current study [8, 33, 34]. This highlights the need for clinicians to initiate cost-transparent discussions with families when planning assessments and interventions, tailoring their discussions to families' circumstances, and identifying avenues for social support, such as referral to a social needs' navigator [35].

### Limitations and strengths

This study had several strengths and limitations. The involvement of research partners with lived experience enhanced the relevance and validity of the survey design and interpretation of results. The use of mixed-methodologies enabled a holistic interpretation of findings [18]. Nonetheless, this study has some limitations. Qualitative data revealed that families incurred additional disability costs which were not explicitly included in the survey, meaning our cost estimates may be an under-estimation. This study also did not consider the significant emotional or time costs associated with caring for a person with CP, the later has been addressed in a previous Australian study and found total opportunity cost for informal care to be 2.36 billion AUD [2]. As participants self-selected to participate, selection biases may be present. The survey was also only available in English, meaning our findings may not be generalizable to culturally and linguistically diverse families, including some who were ineligible to receive NDIS funding at the time this study was conducted [36, 37]. The study findings are also context dependent, relevant to the Australian economic climate during the COVID-19 pandemic in 2020–2021. Future research could investigate whether ability to self-manage NDIS funding plans influences out-of-pocket costs. It may also be helpful to further break down out-of-pocket costs for smaller age brackets to identify when the period of greatest financial need is.

## Conclusion

Almost all individuals with CP and their families incur a wide range of out-of-pocket costs across the lifespan and may experience financial distress, regardless of disability severity. Out-of-pocket cost estimates for CP may empower individuals and families with knowledge to assist with financial planning and budgeting [3]. These findings can be used to advocate for equitable access to support, thus easing potential financial distress. Cost estimates and financial distress indicators should be used to inform government policy, funding and clinical decisions when planning treatments and interventions to best support people affected by CP and their families [3].

## Supporting information

**S1 Table. Checklist for Reporting of Survey Studies (CROSS).**
(DOCX)

**S2 Table. Adult sociodemographic characteristics.**
(DOCX)

**S3 Table. Percentage of participants who would like to spend more across cost types and GMFCS Level.**
(DOCX)

**S1 Appendix. Financial costs survey for families and people living with cerebral palsy in Australia.**
(PDF)

## Author Contributions

**Conceptualization:** Annabel Webb, Claire Galea, Isabelle Balde, Sophie Marmont, James Espie, Nadia Badawi, Sarah McIntyre.

**Data curation:** Annabel Webb, Isabelle Balde, Sarah McIntyre.

**Formal analysis:** Georgina Henry, Annabel Webb, Sarah McIntyre.

**Funding acquisition:** Alison Pearce.

**Investigation:** Alison Pearce, Isabelle Balde, Fiona Garrity, Sophie Marmont, James Espie, Sarah McIntyre.

**Methodology:** Claire Galea, Alison Pearce, Isabelle Balde, Fiona Garrity, Sophie Marmont, James Espie, Sarah McIntyre.

**Project administration:** Isabelle Balde, Sarah McIntyre.

**Resources:** Alison Pearce.

**Supervision:** Alison Pearce, Nadia Badawi, Sarah McIntyre.

**Validation:** Fiona Garrity, Sophie Marmont, James Espie.

**Writing – original draft:** Georgina Henry, Sarah McIntyre.

**Writing – review & editing:** Georgina Henry, Annabel Webb, Claire Galea, Alison Pearce, Isabelle Balde, Fiona Garrity, Sophie Marmont, James Espie, Nadia Badawi, Sarah McIntyre.

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
