## [Decision Letter · Decision Letter 0]

15 May 2023

PONE-D-23-09939

Out-of-pocket costs for families and people living with cerebral palsy in Australia

PLOS ONE

Dear Dr. Henry,

Thank you for submitting your manuscript to PLOS ONE. After careful consideration, we feel that it has merit but does not fully meet PLOS ONE’s publication criteria as it currently stands. Therefore, we invite you to submit a revised version of the manuscript that addresses the points raised during the review process.

Please submit your revised manuscript by  Jun 29 2023 11:59PM. If you will need more time than this to complete your revisions, please reply to this message or contact the journal office at plosone@plos.org. Please include the following items when submitting your revised manuscript:

We look forward to receiving your revised manuscript.

Kind regards,

Josue Mbonigaba, Ph.D

Academic Editor

PLOS ONE

Journal Requirements:

3. We note you have included a table to which you do not refer in the text of your manuscript. Please ensure that you refer to Table 5 in your text; if accepted, production will need this reference to link the reader to the Table.

Reviewers' comments:

Reviewer's Responses to Questions

**Comments to the Author**

1. Is the manuscript technically sound, and do the data support the conclusions?

Reviewer #1: Yes

Reviewer #2: Partly

2. Has the statistical analysis been performed appropriately and rigorously? 

Reviewer #1: Yes

Reviewer #2: I Don't Know

3. Have the authors made all data underlying the findings in their manuscript fully available?

Reviewer #1: No

Reviewer #2: No

4. Is the manuscript presented in an intelligible fashion and written in standard English?

Reviewer #1: Yes

Reviewer #2: Yes

5. Review Comments to the Author

Reviewer #1: The authors are to be congratulated for researching such an important and interesting topic that will greatly interest the journal's readership.

The paper is well-written and presented with clarity. However, while it makes an essential contribution to the literature, some recommendations are outlined below to improve the paper.

ABSTRACT:

1. I suggest adding the study designed after the study aimed. For example,' The study was a cross-sectional survey using mixed-methods of quantitative and qualitative approaches.'

2. How many respondents responded to the open-ended questions?

INTRODUCTION:

1. It is recommended that the authors include a paragraph outlining the definition of out-of-pocket cost.

METHODS

1. It would be helpful to the reader to include a copy of the instrument in the appendix.

2. Provide a brief reliability section that refers to the pilot study conducted as stated in the abstract.

3. Additionally, provide a brief statement about the trustworthiness of the responses to open-ended questions (qualitative findings).

RESULTS:

1. Please check and revise the table number in the text and the description. For example:

A comprehensive breakdown of median costs for thirty different cost subtypes 257 (e.g. neurologist, allied health, rehabilitation specialist) by age group can be seen in Table 3.

Illustrative quotes can be seen in Table 3.

Table 3. Annual Costs Across Sub-types.

2. Please provide the table of regression analysis results for better interpretation and understanding of the results by the reader.

DISCUSSIONS

1. Please introduce a sub-item: 'Limitations and strengths of the study."

CONCLUSION

1. Include a conclusion section in this paper, as it missed this conclusion section.

Thank you. Wishing you all the best.

Reviewer #2: Thank you for the opportunity to review this manuscript reporting an important and timely research examining out-of-pocket costs for families and people living with cerebral palsy in Australia. I applaud the thorough inclusion of people living with cerebral palsy, which is clearly evident in the study design and no doubt, has added great value to the relevance and applicability of the findings to clinical practice and policy.

Please find my comments below.

Methods

I don’t think the methods used for the qualitative data fall within the Framework approach, which applies a pre-designated framework (designed prior to data collection) to the analysis. For this study, the authors describe formulating a coding tree (framework) through reading and initial analysis of the data, and then meeting and agreeing on this coding tree, including discussing inconsistencies. This is a key step in thematic analysis (see Braun and Clarke).

Results

It is interesting to see that 54% of adults who completed the survey via self-report earned less than $31,199 per year. This is much higher when compared with carers who completed the survey. How do these findings compare with the general population (census data)? Do they indicate a level of poverty in this group of people living with CP? This is important to acknowledge and understand in developing health and social care policy.

I pre-empt the following comments with the declaration that my expertise does not lie with statistics, although I have a fair working knowledge of regression analyses. First, I would like to see a table presenting the regression analyses results. This is essential data that the reader needs to be available.

Second, I’m not sure if it is correct to refer to the independent variables as “predictors” in this study. I believe that associations have been established, although this is quite different to a prediction model. I had a brief look at the literature and found this article about association versus prediction interesting: https://www.sciencedirect.com/science/article/pii/S0168822720307518

I am sure the journal will have the results checked by a statistical expert, who will be able to clarify the above.

Discussion

The use of mixed methods is a strength of this research and I like the way the results are integrated in the discussion.

Additional challenges are faced by people who are renting and/or cannot afford a car and hence are not eligible for funding for modifications. Is this reflective of the inverse care law, where those who are most disadvantaged (and in need) are least able to access the care they need?

It is interesting to see that overall, costs seem to be highest in the age 7 – 17 group. Perhaps due to the large growth and associated changing needs during this period. It would be interesting to break this down (in future research) to elucidate when the period of greatest need is. My initial thought is that it may be during the teenage years, with so many varied adjustments to make. Such a breakdown might be of great value for policy makers.

Limitations

An important limitation to mention is that the survey was very long (144 questions over 36 pages), which resulted in a 78% completion rate. In light of this, the authors should consider how the length of the survey might be reduced to optimise completion in future research.

Is it relevant to include a limitation related to the timing of the survey (August 2020 – February 2021), which was during the Covid-19 pandemic and at a time when many consultations would have been virtual, possibly attracting lower consultation fees? Just something for the authors to consider …

6. PLOS authors have the option to publish the peer review history of their article (what does this mean?). If published, this will include your full peer review and any attached files.

Reviewer #1: No

Reviewer #2: No

---

## [Author Response · Author response to Decision Letter 0]

28 Jun 2023

Dear Dr. Emily,

Thank you for the opportunity to make revisions to our manuscript. We have addressed the following journal requirements and below is the response to the reviewers.

Please ensure that your manuscript meets PLOS ONE's style requirements

Response: File names have been updated in accordance with PLOS ONE’s style requirements. 

In your Data Availability statement, you have not specified where the minimal data set underlying the results described in your manuscript can be found

Response: After very helpful discussions with our Human Research Ethics Committee at the University of Sydney, we are unable to make the raw data publicly available. We do not have participant consent to do this, and it is potentially re-identifiable. Please see our updated data availability statement: “Participants did not consent to making their data publicly available and the raw data of this study includes sensitive information and indirect identifiers which may risk the identification of participants. All data requests can be sent to The University of Sydney Human Research Ethics Committee human.ethics@sydney.edu.au.”

We note you have included a table to which you do not refer in the text of your manuscript.

Response: That was an error. Thank you, all Tables are now referred to in the text of the manuscript.

Please review your reference list to ensure that it is complete and correct

Response: All references have been reviewed and completed appropriately in accordance with PLOS One submission guidelines. No existing references have been retracted. Additional references have also been added to address some of the reviewers’ suggestions. 

Abstract

Reviewer 1

I suggest adding the study designed after the study aimed. For example,' The study was a cross-sectional survey using mixed-methods of quantitative and qualitative approaches.'

Response: Thank you for your suggestion – we appreciate your interest in the qualitative components of this study. This study was originally designed as a quantitative survey, rather than a mixed-methods study. However, due to the (unexpectedly) large number and detail of responses to open-ended questions, this data was analysed qualitatively as guided by the literature. We have adapted your suggestion and described the study design in the abstract as “A cross-sectional quantitative survey design was used with qualitative approaches to analyse open-ended questions” (lines 37-38). 

How many respondents responded to the open-ended questions?

Response: Open-ended questions were completed by 231 (85%) participants. We have included this at the beginning of the results section (line 194) as we do not have the word count to include it in the abstract. 

Introduction

Reviewer 1 

It is recommended that the authors include a paragraph outlining the definition of out-of-pocket cost

Response: We have included two additional sentences defining out-of-pocket costs at the beginning of paragraph 3 of the introduction - “Out-of-pocket costs refer to financial expenses which families and individuals pay for using their own income. These expenses are typically either not covered, or only partly covered, by government funding or insurance schemes” (lines 76-78). 

Methods

Reviewer 1 

It would be helpful to the reader to include a copy of the instrument in the appendix.

Response: A copy of the instrument has been uploaded as S1 Appendix. 

Provide a brief reliability section that refers to the pilot study conducted as stated in the abstract.

Response: The survey was co-designed with our research partners and was not formally piloted to generate reliability data. We have amended the abstract to state “co-designed with people with lived experience” to describe the survey development process (line 39). 

Additionally, provide a brief statement about the trustworthiness of the responses to open-ended questions (qualitative findings).

Response: A brief statement has been added to describe the trustworthiness of the qualitative findings: “Researcher triangulation were adopted to increase the trustworthiness of the qualitative analysis – all analyses were reviewed and discussed among our multi-disciplinary team of investigators, including our research partners with lived experience of CP. Participants across all subgroups (age and GMFCS levels) responded to open-ended questions, indicating the case-to-case transferability of the data” (lines 187-191). 

Reviewer 2 

I don’t think the methods used for the qualitative data fall within the Framework approach, which applies a pre-designated framework (designed prior to data collection) to the analysis. For this study, the authors describe formulating a coding tree (framework) through reading and initial analysis of the data, and then meeting and agreeing on this coding tree, including discussing inconsistencies. This is a key step in thematic analysis (see Braun and Clarke).

Response: Thank you for your feedback. Our readings identified that the Framework approach is a type of thematic analysis which can be used either deductively or inductively (Gale et al. 2013). We have now specified this and have referenced Gale et al. 2013 in our paper for greater clarity (see lines 181-182). Framework analysis has six stages which we have used: 1) Transcription; 2) Familiarisation of data; 3) Develop coding framework; 4) Indexing; 5) Charting; 6) Mapping and interpretation. These stages are very similar to the six offered by Bruan & Clarke (2006) - Framework analysis sits within the broad family of “thematic analysis” methods (Gale et al. 2013). We have also included the Bruan & Clarke (2006) reference to acknowledge their contribution to this type of methodology. 

Results

Reviewer 1 

Please check and revise the table number in the text and the description

Please provide the table of regression analysis results for better interpretation and understanding of the results by the reader

Response: A table of regression analysis results has been included (line 278; page 16). All table numbers and descriptions have been amended and updated accordingly. 

Reviewer 2 

interesting to see that 54% of adults who completed the survey via self-report earned less than $31,199 per year. This is much higher when compared with carers who completed the survey. How do these findings compare with the general population (census data)? Do they indicate a level of poverty in this group of people living with CP? This is important to acknowledge and understand in developing health and social care policy.

Response: Thank you for this suggestion – we agree this is important to acknowledge. We feel this would be most appropriate to include in the discussion and have included the following text “This is reflected in the majority (54%) of adults with CP who self-reported earning less than $31, 199 p.a.. This indicates highly disproportionate levels of financial poverty, as compared to 21% of the Australian adult population during 2019-2020.” (lines 391-394).

First, I would like to see a table presenting the regression analyses results. Second, I’m not sure if it is correct to refer to the independent variables as “predictors” in this study.

Response: A table of regression analysis results has been included (line 277; page 16). We have also updated the language to describe all regression results to clarify that associations have been established (see lines 46, 53-54, 219-221, 228-229, 247, 305-306 and 377). 

Discussion 

Reviewer 1 

Please introduce a sub-item: 'Limitations and strengths of the study."

Response: We have introduced a sub-item “Limitation and Strengths” on line 411.

Include a conclusion section in this paper, as it missed this conclusion section

Response: We have included a conclusion subheading on line 429. 

Reviewer 2

Additional challenges are faced by people who are renting and/or cannot afford a car and hence are not eligible for funding for modifications. Is this reflective of the inverse care law, where those who are most disadvantaged (and in need) are least able to access the care they need?

Response: Thank you for this suggestion – we agree this is reflective of the inverse care law. We have incorporated this into the discussion by explaining that “Participants also reported additional challenges accessing funding when experiencing socioeconomic disadvantage, for example those who seemed to be ineligible for home or vehicle modifications because they were unable to afford their own home or a relatively new car. This finding is reflective of the inverse care law, whereby those who deserve the most support, are least able to access the care they need” (lines 358-362). 

It is interesting to see that overall, costs seem to be highest in the age 7 – 17 group. Perhaps due to the large growth and associated changing needs during this period. It would be interesting to break this down (in future research) to elucidate when the period of greatest need is. My initial thought is that it may be during the teenage years, with so many varied adjustments to make. Such a breakdown might be of great value for policy makers.

Response: We agree, this is an interesting and important point for further research. We have acknowledged this in the discussion as an avenue for future research “It may also be helpful to further break down out-of-pocket costs for smaller age brackets to identify when the period of greatest financial need is” (lines 426-428). 

An important limitation to mention is that the survey was very long (144 questions over 36 pages), which resulted in a 78% completion rate. In light of this, the authors should consider how the length of the survey might be reduced to optimise completion in future research.

Response: Thank you for this consideration. We acknowledge the final survey was long, but the survey was codesigned with people who have lived experience of CP and who are most knowledgeable about what out-of-pocket costs directly affect families. Whilst drafting iterations of the survey, our research partners continually thought of additional out-of-pocket costs they have – all of which were important to include in the survey to obtain accurate out-of-pocket cost estimates. We did our best to streamline and limit, however our feedback from our research partners was that it wasn’t too long and they didn’t want any sections removed. This was reflected in our research findings, whereby total out-of-pocket costs spanned across each of the nine cost types and thirty subtypes. Furthermore, participants reported they had several additional costs in their open-ended responses. Hence, despite our best efforts to comprehensively capture all disability-related costs, our findings are probably still an underestimation (as stated in our discussion lines 415-417). 

Is it relevant to include a limitation related to the timing of the survey (August 2020 – February 2021), which was during the Covid-19 pandemic and at a time when many consultations would have been virtual, possibly attracting lower consultation fees? Just something for the authors to consider

Response: This was an important point of consideration when disseminating the survey. We were very conscious to distribute the survey as quickly as possible in early 2020 to best try to minimise the impact of the COVID-19 pandemic on survey responses. The COVID-19 pandemic may have influenced out-of-pocket costs for families in different ways. This is especially the case given different Australian states were subject to very different restrictions at different times throughout the pandemic. We have clarified this limitation by inserting “during the COVID-19 pandemic” when discussing the context-dependent nature of our findings (lines 424-425). 

We hope this addresses the reviewers’ suggestions. 

Warm regards, 

Georgina Henry

---

## [Editor Report · Decision Letter 1]

6 Jul 2023

Out-of-pocket costs for families and people living with cerebral palsy in Australia

PONE-D-23-09939R1

Dear Authors

We’re pleased to inform you that your manuscript has been judged scientifically suitable for publication and will be formally accepted for publication once it meets all outstanding technical requirements.

Kind regards,

Josue Mbonigaba, Ph.D

Academic Editor

PLOS ONE
---

## [Editor Report · Acceptance letter]

11 Jul 2023

PONE-D-23-09939R1 

Out-of-pocket costs for families and people living with cerebral palsy in Australia 

Dear Dr. Henry:

I'm pleased to inform you that your manuscript has been deemed suitable for publication in PLOS ONE. Congratulations! Your manuscript is now with our production department. 

Kind regards, 

on behalf of

Dr. Josue Mbonigaba 

Academic Editor

PLOS ONE